

# Highly-resolved satellite remote sensing based land-use change inventory
# yields weaker surface albedo-induced global cooling
**Authors:** Xiaohu Jian[1], Xiaodong Zhang[1,*], Xinrui Liu[1], Kaijie Chen[1], Tao Huang[2], Shu Tao[1],
Junfeng Liu[1], Hong Gao[2], Yuan Zhao[2], Ruiyu Zhugu[1], Jianmin Ma[1]
**Affiliations:**
[1]Laboratory for Earth Surface Processes, College of Urban and Environmental Sciences,
Peking University, Beijing 100871, PR China
[2] Key Laboratory for Environmental Pollution Prediction and Control, Gansu Province,
College of Earth and Environmental Sciences, Lanzhou University, Lanzhou 730000, PR
China
[*]**Corresponding author:** Xiaodong Zhang (zhangxd2020@pku.edu.cn)
**Abstract**
Land-use change (LUC) is ranked as the second anthropogenic source of climate change after
fossil fuel burning and yields negative albedo-induced radiative forcing (ARF). This cooling
effect has been assessed using low spatiotemporally resolved LUC datasets derived from
historical statistical data with large uncertainties. Herein, we implement a satellite remote
sensing derived highly resolved LUC dataset into a compact earth system model and reassess
the global and regional surface ARF by LUC from 1983 to 2010 relative to 1750. We find that
the magnitude of negative ARF obtained from the present study is lower by 20% than that
estimated by the Intergovernmental Panel on Climate Change, implying a weaker cooling
effect. The result reveals that the global LUC-induced surface albedo change may not
significantly slow down global warming as was previously anticipated. Sub-Saharan Africa
made the largest net contribution to the magnitude of global ARF (39.2%), due to substantial
land use conversions, typically the conversion from forest to other vegetation lands, which
accompany with higher surface albedos. The most remarkable land cover changes occurred in
East and Southeast Asia, which dominated the changes in global ARF in recent decades. Based



on major land cover types in these two regions, we infer that vegetation lands exert a most vital
effect on global ARF variation.

## 1. Introduction

Anthropological activities that have effectuated global climate change can be primarily

categorized under greenhouse gas emissions, the emissions of aerosols, and land use change
(LUC) (IPCC AR6, 2021). LUC in different temporal and spatial scales varies rapidly from
local to global scales, with significant ramifications for the climate system, and is one of the
key drivers of global climate change (Feddema et al., 2005; Cai et al., 2004; Foley et al., 2005;
Houghton et al., 2012; Zhu et al., 2019). LUC accounts for 13%–20% of the total anthropogenic
carbon emissions from the 1990s to the 2010s, and 20% in the 1980s and 1990s (Houghton et
al., 2012), ranking as the second source of anthropogenic climate change after fossil fuel
combustion (Andrews et al., 2017). The influence of LUC on climate change is primarily
manifested in two critical processes: the radiation/energy interface between the surface and the
atmosphere and the changes in the carbon source/sink. LUC affects climate by emitting or
absorbing greenhouse gases in the atmosphere, modifying the carbon cycle within the climate
system. LUC also modifies the albedo and roughness of the underlying surface, altering the
surface heat budget. By functioning as a carbon sink through carbon reduction-oriented land
management, LUC plays a pivotal role in the sequestration of carbon (IPCC AR6, 2021). Such
LUC-induced carbon sinks are crucial for compensating emissions from other carbon sources,
such as fossil fuel energy, transportation, and housing, that continue to emit carbon dioxide.

The extent of the influence of LUC on the climate system and energy balance is often

measured in terms of radiative forcing (RF) (Andrews et al., 2017; Andrews et al., 2020;
Ramanathan et al., 1975; Bonan et al., 2008; Betts et al., 2000; Ward et al., 2014). The primary
effect of RF on climate change is through a temperature feedback mechanism (Sherwood et al.,
2015). While the effects of LUC on climate balance have been extensively studied (Foley et
al., 2005; Houghton et al., 2012; Vose et al., 2004; Gries et al., 2019), knowledge gaps still
remain in the understanding of LUC-induced climate forcing. This is partly due to the lack of
extensive investigations and uncertainties in this field (IPCC AR6, 2021). The commonly held



belief is that the change in surface albedo associated with LUC has a negative forcing globally, leading to a cooling effect and functioning as a carbon sink. However, the magnitudes of negative forcing vary between $-0.15$ W m$^{-2}$ and $-0.6$ W m$^{-2}$ in different studies spanning the pre-industrial to industrial era (IPCC AR3, 2001; Myhre et al., 2003; Hansen et al., 2004; Betts et al., 2007; Forster et al., 2007; Pongratz et al., 2009; Ward et al., 2014; Li et al., 2016; Jiao et al., 2017). The Intergovernmental Panel on Climate Change (IPCC) AR3 report (2001) (IPCC AR3, 2001) adopted $-0.25 \pm 0.25$ W m$^{-2}$ as the global average RF due to surface albedo change. This value has been revised in subsequent reports to $-0.15 \pm 0.10$ W m$^{-2}$ (IPCC AR6, 2021). The magnitude of negative RF induced by surface albedo (hereafter referred to as ARF) obtained from other studies appears to be greater than the IPCC adopted value (Fig. 1). In AR3 of the IPCC, the scientific understanding of LUC-induced ARF was deemed "very low". Due to the limited number of studies and the uncertainty of historical land cover (LC) changes, IPCC AR6 (2021) assigns these values a medium confidence level. A substantial proportion of the uncertainties in LUC and ARF can be attributed to the lack of high spatiotemporal resolution in LUC data and sufficient supports by measurements (Gong et al., 2013; Winkler et al., 2021; Jian et al., 2022). Recently, numerous high-resolution remote sensing datasets have been used to develop highly resolved LUC datasets (Gong et al., 2013; Winkler et al., 2021). Modern satellites are equipped with sensors that offer high spatial resolution, allowing for the detailed mapping of land-use changes. This level of detail is essential for identifying specific types of land-use changes, such as deforestation, urban expansion, or agricultural intensification, each of which has different impacts on radiative forcing. These remote sensing-based datasets reveal that LUC has affected as much as one-third of the world's land area in just six decades (1960–2019), roughly four times greater than the estimates from long-term land change assessments conducted previously (Winkler et al., 2021). It is interesting to know if and to what extent recently developed remote sensing-based global land use (LU) change data with very high spatial-temporal resolution from a climate perspective and potentially low uncertainty could improve the estimation of LUC-induced global and regional climate forcing.

In the present study, we reassessed the LUC forced ARF by incorporating a high-resolution (5 km×5 km) satellite remote sensing measured LUC dataset into a compact earth



system model (see Methods) and evaluate the contributions from various LUC and LU types
in different regions/countries to global ARF, aiming to provide a more precise and
measurement-based estimate of regional and global ARF.

**2. Materials and Methods**
**2.1. OSCAR Model**
OSCAR v2.4 (Gasser et al., 2017), a compact model of global biogeochemical cycles, is
used to investigate the effect of LUC-induced changes in surface albedo on global RF. OSCAR
is not spatially resolved but country and region-based. It is a nonlinear box model incorporating
as many key climate components and modules as possible, such as LU change and aerosol
physics-chemistry feedback. The model was designed to simulate long-term trends in earth
system change rather than seasonal and interannual variations in the earth system. OSCAR is
also a parametric model in which several parameters required to calculate RF are calibrated on
(or input from) complex climate models. Model uncertainties are assessed by Monte Carlo
ensembles. In the present study, we have assigned a 5% uncertainty in OSCAR modeled ARF
based on LUC data uncertainty. Further details, advantages of OSCAR model, and the
motivations to use OSCAR model in our ARF simulations are presented in Supplementary Text

1.


**2.2. Updated Global LUC Data**
The OSCAR's capability to simulate LU change-induced RF is one of its strengths. To
assess the combined effects of human activities on the carbon-climate system (Hurtt et al.,
2011), the model employs the LU Harmonization (LUH1) LUC dataset developed under IPCC-
AR5. The results show a smooth transition of annual changes in LUC, suggesting that approach
and data sources adopted to derive LUH1 (Supplementary Text 2) likely missed some
important characteristics of LU transitions, resulting in a substantial uncertainty in the modeled
LUC-induced RF. Although LUH1 was recently updated to LUH2 with a spatial resolution of
$0.25° \times 0.25°$ latitude/longitude (Hurtt et al., 2020), in the present study, we chose the Global
Land Surface Satellite-Global LC dataset (GLASS-GLC) (Liu et al., 2020) to replace the LUH1



inventory with coarse spatial resolution in the OSCAR model to capture the temporal-spatial variations of LUC adequately. GLASS-GLC was developed using 5 km×5 km resolution GLASS (Global Land Surface Satellite) climate data records from 1982 to 2015. Although both LUH and GLASS-GLC provide annual LUC, compared to previous LUC products, such as LUH1 and LUH2, GLASS-GLC based on satellite remote sensing has greater consistency, a higher spatial resolution, and many LU types. Compared to LUH1 dataset derived based on historical statistics and census data combining with the History Database of the Global Environment (HYDE) model and the Global Land-use Model (GLM) (Hurtt et al., 2011), the GLASS-GLC dataset uses the Google Earth Engine (GEE) platform with the latest version of GLASS CDRs (climate data records) from 1982 to 2015 (Liu et al., 2020) to obtain a more reliable land use inventory. GLASS-GLC considers seven LUC classes, including cropland, forest, grassland, shrubland, tundra, barren land, and snow/ice, with an overall accuracy of 82.81%. Although the GLASS-GLC data source also include urban areas, these small areas are not straightforward to be distinguished at the 5 km×5 km resolution as compared to other LUCs (Liu et al., 2020). Although urban expansion could contribute to climate warming (Ouyang et al., 2022), our previous work (Jian et al., 2022) has explored the impact of urbanization on China's ARF and found that the impact of urban sprawl on China's ARF is very small (0.59%) and hence can be neglected, although China has experienced the world's most rapid urbanization since the 1980s, due to considerably smaller area of urban land than the other selected 6 LU categories. Likewise, the urban land also exerts a little effect on ARF from a global perspective. Therefore, urban areas were not taken into consideration in this study. The LUC data are available for download at https://doi.org/10.1594/PANGAEA.913496. Noted that, although the updated GLASS-GLC was extended to 2015, given that some of parameters and variables in OSCAR v2.4 were only available up to 2010, we performed OSCAR simulations from 1982 to 2010.

The surface roughness affects primarily on turbulent exchange of heat and air mass between the underlying surface and air, which may indirectly alter surface radiation fluxes via changing sensible and latent fluxes under a heat balance status (Andrews, 2012). Since the OSCAR model does not consider the surface roughness length, the impact of LUC on surface



roughness is not included in the present study.

**2.3. Sensitivity Analysis**
To illustrate the influence of LUC-induced albedo change on the global RF, we chose
five LU types that have dominated the global LUCs over the past four decades: cropland, desert,
forest, grassland, and shrub. We carried out extensive sensitivity experiments by reducing each
LU transition area by 20% within five major LU types (cropland, desert, forest, grassland, and
shrub), aiming to examine the relative significance and contribution the LU conversion and
transition among different LU types to the ARF. Among them, each LU type is converted to
the rest four LU types, thereby accounting for total 20 LU transitions and sensitivity
experiments. However, in the original OSCAR inventory, there were only inter-conversions
between cropland and other land types, and no conversions between desert, forest, grassland,
and shrub. Table S1 presents these 20 LU transitions from 1982 to 2010. To facilitate analysis
and refine the effect of LUC on ARF, the world has been divided into nine regions. These
regions include East and Southeast Asia (including China), Europe, Latin America, the Near
East and North Africa, North America, Oceania, Russia, Sub-Saharan Africa, and South Asia
(Fig. 2). Table S2 presents the surface albedos for the five LU types in each nation and the nine
regrouped global regions. Between the OSCAR LUH1-LUC inventory and the GLASS-GLC
inventory, Fig. S1 and Table S3 compare annual changes in the area of each LU type from
1982 to 2010 in the globe and the nine regions. There are distinct differences between the two
LUC inventories. The causes of these differences and two simulation results are discussed in
Supplementary Text 2. By performing OSCAR simulations with a low spatiotemporally
resolved OSCAR LUH1-LUC inventory (Scenario 1) and a high spatiotemporally resolved
GLASS-GLC inventory (Scenario 2), respectively, we also set up two model scenarios for
sensitivity experiments.

**2.4 Methods of comparing ARF results for two datasets**
The percentage changes in annual ARF between the two scenarios are estimated using the
following equation:



$$ARF_F = (ARF_{S2} - ARF_{S1}) \times 100\% / ARF_{S1} \tag{1}$$


where $ARF_F$, $ARF_{S1}$, and $ARF_{S2}$ represent the percentage changes in ARF and ARF values from
model scenarios 1 and 2, respectively.

**2.5. Disturbance Capacity Analysis and Effective Area**

We conducted comprehensive sensitivity experiments on OSACR simulations to analyze

the impact of each of the 20 LU conversions on ARF globally and across of the nine regions.
We consider the conversion from each of the five LU types to the remaining four LU types,
resulting in 20 LU conversion types (Table S1). In these sensitivity experiments, we introduce
a disturbance capacity (DC, %) that determines the magnitude of the ARF change induced by
the 20 LU conversions in the region of interest. The DC is defined as follows:
$$
\begin{cases}
\Delta RF_{ij} = \overline{RF_i} - \overline{RF'_{ij}}, \\
\\
DC_{ij} = \dfrac{\Delta RF_{ij}}{\sum_{j=1}^{20} |\Delta RF_{ij}|} \ * \ 100\%.
\end{cases}
\tag{2}
$$

where, $\overline{RF_i}$ represents mean ARF in region $i$ averaged from 1983 to 2010. We reduce the $j^{th}$
LU conversion in region $i$ by 20% and define resulted ARF in region $i$ as $RF'_{ij}$ in each year.
Its mean from 1983 to 2010 is defined as $\overline{RF'_{ij}}$. Expression (2) can also be considered as a
statistical formula for determining the relative significance or the contribution of ARF induced
by a particular LU conversion to the total ARF change across all regions and LU conversion
types. For example, the sensitivity experiment for grassland to cropland conversion in region $i$
(13$^{th}$ sensitivity experiment or LU conversion) was conducted by multiplying the area
converted from grassland to cropland by 0.8, indicating a 20% reduction in the grassland to the
cropland transition area. The changes (or response) of ARF in region $i$ perturbed by a 20%
reduction in the $j^{th}$ LU conversion area $\Delta RF_{ij}$ were then used to estimate $DC_{ij}$ (Eq. 2).

We also examine net LU conversion among the five LU types, where net LU conversion

is defined as the difference between a pair of LU conversions. For instance, the net conversion
from grassland to cropland (13$^{th}$ LU conversion, Table S1) and from cropland to grassland (3rd



LU conversion, Table S1) is calculated as the area converted from grassland to cropland minus
the area converted from cropland to grassland, also referred to as the net two-way conversion.
This adjustment reduces the total LU conversions in the sensitivity experiment from 20 to 10.
The DC for the ten net LU conversion areas is definable as follows:

$$
\begin{cases}
A_{a\leftrightarrow b}^{t} = A_{a\rightarrow b}^{t} - A_{b\rightarrow a}^{t}, \\
\\
DC_{a\leftrightarrow b} = \dfrac{DC_{a\rightarrow b}}{|DC_{a\rightarrow b}|} * \left( \dfrac{|DC_{a\rightarrow b}| + |DC_{b\rightarrow a}|}{2} \right).
\end{cases}
\tag{3}
$$

where $A_{a\leftrightarrow b}^{t}$ is the area of net LU transition, $a$ and $b$ indicate the conversion from LU type $a$
to type $b$, respectively, $A_{a\rightarrow b}^{t}$ and $A_{b\rightarrow a}^{t}$ are the transition areas from LU type $a$ to LU type $b$,
and from LU type $b$ to $a$. The superscript $t$ denotes a specific year between 1982 and 2010.
$DC_{a\leftrightarrow b}$ represents the disturbance capacity of net conversion between paired LUs, and $DC_{a\rightarrow b}$
and $DC_{b\rightarrow a}$ are the DC of LU conversion $a \rightarrow b$ and $b \rightarrow a$, respectively. After the DC of LU
conversion is determined, we estimate an effective area (EA) (see main text), which is defined
here as the cumulative area of six net LU conversions, given by

$$
\begin{cases}
\alpha_{ik} = \dfrac{DC_{ik}}{\sum |DC_{ik}|}, if \ |DC_{ik}| \geq 1\%, \\
\\
\alpha_{ik} = 0, if \ |DC_{ik}| < 1\%, \\
\\
A_{it}^{e} = \sum_{k=1}^{10} \alpha_{ik} * A_{ikt}.
\end{cases}
\tag{4}
$$

where $DC_{ik}$ represents the DC in the *kth* net LU conversion type affecting ARF in region $i$.
The ratio of $DC_{ik}$ to the absolute value of total $DC_{ik}$, defined by in Eq. (4), can also be viewed
as the proportion of different net LU conversions to the global EA (Table S4). $A_{it}^{e}$ denotes the
EA in year $t$ and region $i$. $A_{it}^{e}$ indicates the area of the *kth* net LU conversion type in year $t$
and region $i$. Consequently, the EA measures the extent of a LU conversion area that
significantly impacts the change in ARF. In calculating the EA, we first exclude net LU
conversions with |DC| > 1%, then sum up these |DC| values (Σ|DC|) and finally divide the DC





of each net LU conversion with |DC| > 1% by Σ|DC| (Eq. 4). Table S4 presents the correlation
coefficients and significance tests of the EAs. According to Eq. 4, once the DC is obtained, the
EA area can be estimated, which defines the converting areas of the 10 net land conversion
types between 1982 and 2010 divided by their respective absolute DCs. The results explain the
change in ARF from 1983 to 2010 (Supplementary Text 3).

**2.6. Quantifying the Contribution of Regional LU Transition to Changes in Global ARF**
**and Effective Area**

The changes in ARF due to LU conversion in a region from 1983 to 2010 can be simply

defined as the differences in ARF between 1983 and 2010. First, we considered the ARF
change in any region across the globe as:
$\Delta ARF_{LU\_all} = RF_i'^{\,2010} - RF_i'^{\,1983}$.                                    (5)
where, $RF_i'^{\,2010}$ and $RF_i'^{\,1983}$ denote the ARF in *the $i^{th}$* region in the S2 scenario using the
GLASS-GLC inventory in 2010 and 1983, respectively. To remove the effect of LU conversion
on ARF, we reduced the transition area of each LU type from 20% to 100% in the 20 sensitivity
experiments, meaning no occurrence of LU transition. Second, we introduced
$RF_{i,j}^{2010}$ and $RF_{i,j}^{1983}$ to represent the ARF in the $i^{th}$ region induced by the $j^{th}$ LU transition in
the S2 scenario in 2010 and 1983, so their differences are as follows:
$\Delta ARF_{LU\_ind} = RF_{i,j}^{2010} - RF_{i,j}^{1983}$.                                    (6)
This can be regarded as the changes in ARF induced by other 19 conversion types for the $j^{th}$
LU conversion during this period. The changes in ARF subject to any LU conversion in any of
the nine regions can be written as:
$\delta_F = \Delta ARF_{LU\_all} - \Delta ARF_{LU\_ind}$.                                    (7)



243   In other words, $\delta_F$ indicates the net effect of regional LU transition on ARF. Finally, the

244   contribution of ARF from any region and any LU conversion to the changes in global ARF is

245   defined as:

246   $$C_{ARF} = \frac{\delta_f}{ARF_{global}^{2010} - ARF_{global}^{1983}}.$$            (8)

247   here, $ARF_{global}^{2010}$ and $ARF_{global}^{1983}$ are global ARF in 2010 and 1983 from model scenario 2.

248   Their difference is constant (0.0364 W m$^{-2}$).

249    The contribution of regional EAs to the global EA is simply estimated by Eq. 9:

250   $$C_{EA} = \alpha_{i,k} \times \frac{\sum_{n=1}^{28} \frac{EA_{n,i}}{\sum_i^9 EA_{n,i}}}{28}.$$         (9)

251   where, $i = 1, 2, …9$ denotes nine regions, $n = 1, 2, … 28$ is the number of years from 1983 to

252   2010 and $\alpha_{i,k}$ is defined in Eq. (4).

253    The contribution of two-way LU conversions to the changes in global ARF is defined by

254   Eq. 10:

255   $$C_{LV}^k = \sum_{i=1}^9 \sum_{j=1}^8 C_{ARF}^{i,j}.$$            (10)

256   where, $k = 1,2…5$ denotes five LU types, $i = 1, 2,…,9$ denotes nine regions, and $j = 1,2,…,8$

257   indicate paired two-way LU transitions. Taking cropland as an example, the one-way

258   transitions between cropland to the remaining four LU types are the transitions from cropland

259   to forestland, grassland, desert, and shrubland. The other way includes transitions from

260   forestland, grassland, desert, and shrubland to cropland. So the two-way transition includes

261   eight LU conversions.

263   **3. Results**

264   **3.1. Response of Global RF to Perturbed Albedo**



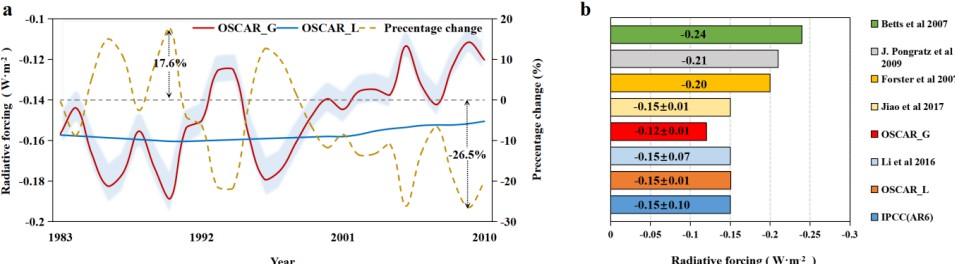

**Figure 1.** Annual RF (W m⁻²) due to albedo change and ARF percentage change (%) and different ARF derived from previous studies. (a) OSCAR-modeled annual RF due to the albedo change and ARF percentage change between the two model scenarios S1 and S2 from 1983 to 2010 derived from coarse resolution LUH1-LUC inventory (solid blue line, see 'Methods') and GLASS-GLC inventory (solid red line, see Methods). The annual ARF of 1983 through 2010 from both model scenarios is relative to the baseline year of 1750. Two-tailed T-Test yields a p-value of 0.025 (<0.05), indicating the statically significant difference between OSCAR-G and OSCAR-L data series. Pale blue shading indicates the uncertainty interval estimated from Monte Carlo simulations. Dashed yellow line stands for a percentage change in ARF from the two scenarios. (b) ARF from present (red color bar) and previous (other color bars) studies from the industrial era to 2010.

To examine the extent of the changes in global RF subject to the altered surface albedo derived from LU transition from 1983 to 2010, we compared the ARF using the coarse resolution LU Harmonization v1-LU Change [LUH1-LUC inventory (OSCAR_L, model scenario S1)] extending from 1750 to 2010 and the fine resolution Global Land Surface Satellite-Global LC dataset [GLASS-GLC inventory (OSCAR_G, model scenario S2)] in OSCAR simulations. It is noted that the annual ARF derived from the model scenario 1 was relative to the baseline year of 1750. The annual ARF derived from the model scenario 2 was also relative to 1750 but we replaced LUH1-LUC with GLASS-GLC after 1982. Fig. 1a depicts the OSCAR-simulated annual global ARF subject to the two model scenarios. From 1983 to 2010, annual ARFs derived from the two LUC scenarios demonstrated upward trend. In contrast, the ARF in the S1 simulation (solid blue line) displays a smoother variation and a weaker increase with a linear trend of 0.0003 (*P*-value < 0.01). The smooth transition from



historical LUC estimates to future projections in the LUH1-LUC results in such gradual
changes in the ARF. In contrast, the ARF in the S2 simulation (solid red line) displays strong
interannual fluctuations and a more rapid increase with a linear trend of 0.0018 ($P$-value <
0.01). The dashed brown line indicates the resulting $ARF_F$ ranges from −26.5% (2009) to 17.6%
(1990). Globally, both scenarios produce negative forcing, consistent with previous estimates
(IPCC AR6, 2021; Li et al., 2016). As aforementioned, even though we only replaced the
coarse resolution LUH1-LUC inventory with the fine resolution GLASS-GLC inventory, the
ARF in the OSCAR is predicted since the industrialization era in the 1750s, the same as the
IPCC AR6. This suggests that both scenario simulations utilized the same LUH1-LUC data
before 1982. Consequently, significant annual and decadal changes in ARF have occurred over
the past few decades, alongside rapid and remarkable global variations in LUC. The significant
differences in the ARF between the two scenarios can be attributed to different data sources
and approaches applied to derive LUH1-LUC and GLASS-GLC. The former was developed
from a combination of historical statistics, population census data, HYDE, and GLM models.
Because the time covered by this inventory are outside the period of satellite observations,
large uncertainties in LUH1-LUC have been recognized (Hurtt et al., 2020). A higher
resolution land use dataset can ensure interannual consistency and comparability of the LUC,
and enables the accurate estimation of the rate and mode in LUC (Gong et al., 2013; Liu et al.,
2020), which can capture more detailed LUC and LU transitions. Recent reports indicate that
the global LUC is four times larger than previously estimated (Winkler et al., 2021). The
differences between the two LUC datasets are shown in Supplementary Text 2, Text 4, Table
S3, and Fig. S1.

Previously estimated global ARF with coarse resolution LUC data has been subject to

several concerns. According to the IPCC Assessment Reports, the global RF induced by LUC
from pre-industrial times to the present due to changes in land albedo is approximately −0.15
± 0.10 W m$^{-2}$, indicating that ARF plays a cooling role (IPCC AR6, 2021). Considering that
radiative forcing is often accumulated from the past, the differences of ARF from the two
inventories occurred mostly in final year, namely 2010. Our OSCAR simulation under scenario
1 using a LUH1-LUC inventory yielded the same negative ARF value of −0.15 W m$^{-2}$ as



reported by the IPCC (Fig. 1b). Using coarse resolution and historical statistics-based LUC data, additional studies have also obtained ARF results with great uncertainties. As depicted in Fig. 1b, all previous studies yielded stronger negative ARFs than the IPCC's estimate, with the negative ARF reaching as low as $-0.24$ W m$^{-2}$ (Betts et al., 2007). However, our estimation subject to scenario 2 yields an ARF of $-0.12 \pm 0.01$ W m$^{-2}$, which is only half of that reported by Betts et al (2007). The result suggests that the global LUC-induced surface albedo change may not be acting as anticipated to slow down the global warming.

## 3.2. Contribution of Regional LUC to Global ARF

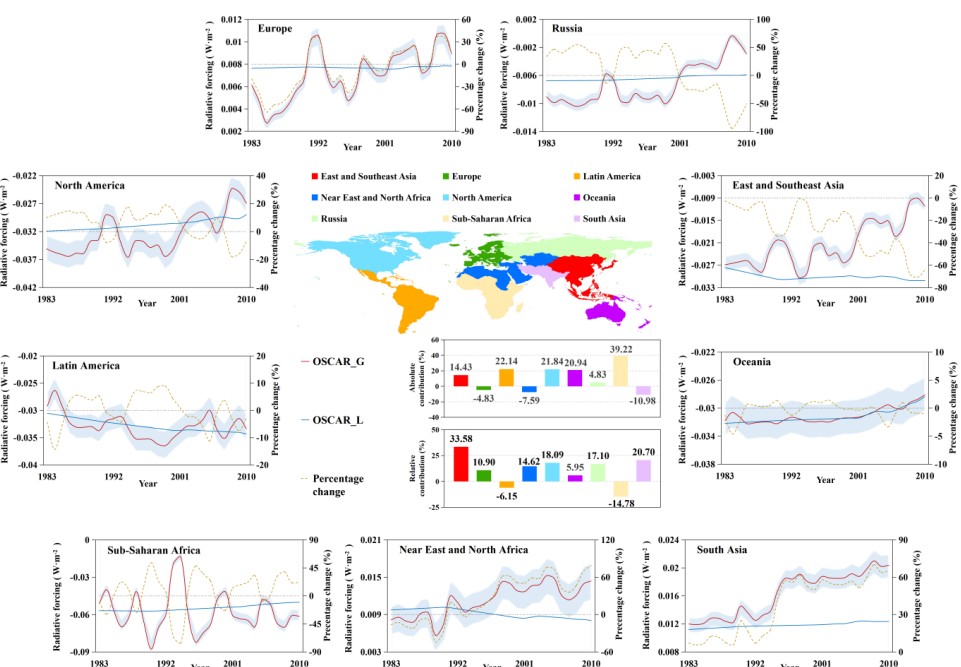

**Figure 2.** Annual RF (W m$^{-2}$) due to the albedo change and ARF percentage change between the two model scenarios S1 and S2 from 1983 to 2010 derived from LUH1-LUC inventory (solid blue line) and GLASS-GLC inventory (solid red line) in nine regions across the globe. The pale blue shading indicates the uncertainty interval estimated from Monte Carlo simulations. The dashed brown line stands for the





percentage change in the annual ARF$_{i-F}$ between the LUH1-LUC inventory (ARF$_{i-S1}$) and GLASS-GLC
inventory (ARF$_{i-S2}$), in which $i$ represents regions, respectively, including East and Southeast Asia, Europe,
Latin America, Near East and North Africa, North America, Oceania, Russia, Sub-Saharan Africa, and South
Asia. The first bar chart illustrates the absolute contribution of different regions to the global albedo-induced
RF, and the second bar chart displays the relative contribution of different regions to the global albedo-
induced RF changes. The nine color bars represent different regions, as indicated by the color legend above
the colored sectional map.

In recent decades, LUC has been subject to significant spatial heterogeneity across the
globe. To investigate the magnitude of the response of global ARF to continental/regional LUC
since the 1980s, we divided 113 countries and regions in OSCAR into nine regions. These
include East and Southeast Asia, Europe, Latin America, the Near East and North Africa, North
America, Oceania, Russia, Sub-Saharan Africa, and South Asia (see Table S2). The colored
sectional map in the center of Fig. 2 indicates the nine regions. In addition, the annual variation
of the ARF subjected to GLASS-GLC (W m$^{-2}$, solid red line, scaled on the left Y-axis) and its
percentage change (%, dashed brown line, scaled on the right Y-axis) in each of these regions
are illustrated in the nine-line charts of Fig. 2. Below the sectional map are two bar charts
depicting the absolute and relative contributions of the nine regions to the global ARF.
Correspondingly, the total contribution is defined as the proportion of the mean ARF of each
nine regions to the global mean ARF from 1983 to 2010. The relative contribution is defined
as the proportion of the change in ARF in each of the nine regions to the change in global ARF
between 1983 and 2010. In addition, the OSCAR-simulated ARFs in each region derived from
LUH1-LUC (solid blue line, scaled to the left on the Y-axis) are displayed in the line charts.
Herein, OSCAR predicts ARFs by incorporating fine-scale variations, as opposed to the LUH1-
LUC-derived ARFs with smoothing variations; GLASS-GLC, on the other hand, displays more
pronounced annual fluctuations. In East and Southeast Asia and Near East and North Africa,
the simulated ARFs derived from the two LUC datasets exhibit opposite trends from 1983 to
2010, indicating that LUC data substantially influence regional and continental ARFs.



As evident from the bar charts below the sectional map, among the nine regions, Sub-
Saharan Africa, with a mean ARF of −0.06 W m$^{-2}$ on average from 1983 to 2010, made the
largest net contribution (39.2%) to the global mean ARF. The significant contribution from
Sub-Saharan Africa is attributable to its large desert area of 697.37 Mha with a high albedo
(Table S2) and pronounced LU conversions among vegetated LU types (Fig. S2 and Table S3).
South Asia had a mean ARF of 0.02 W m$^{-2}$ from 1983 to 2010. This region had an absolute
negative contribution of −10.98% to the global mean ARF averaged over the nine regions, most
likely because of rapidly expanding croplands (226.93 Mha) with low albedo associated with
the Green Revolution (Pingali et al., 2012; Liu et al., 2021; Huang et al., 2022) (Fig. S2 and
Table S3). East and Southeast Asia, Europe, Latin America, Near East and North Africa, North
America, Oceania, and Russia contributed 14.43%, −4.83%, 22.14%, −7.59%, 21.84%,
20.94%, and 4.83% to the global mean ARF, respectively.
Although East and Southeast Asia made a moderate absolute contribution to the global
mean ARF compared to other regions, this region experienced the largest LU change between
1982 and 2010. This was characterized by the highest ARF change (0.017 W m$^{-2}$), comprising
the most significant relative contribution (33.58%) to the global ARF change. Such a
contribution can be attributed to the massive LC changes brought on by afforestation and the
management of land desertification during this time period (Liu & Xin, 2021; Imai et al., 2018;
Zhang et al., 2016), which led to a decrease in surface albedo. In contrast, deforestation in Sub-
Saharan Africa in recent decades (Keenan et al., 2015) promoted rapid shrub growth (Atsri et
al., 2018; Mograbi et al., 2015), resulting in a rise in albedo. Consequently, this region has the
largest negative contribution to the global ARF change, at −14.78%, promoting a cooling effect
on the global climate. Similarly, the deforestation in Latin America caused by the conversion
of forest to cropland and pastureland in recent decades (Armenteras et al., 2021; Hansen et al.,
2013; Nogueira et al., 2019; Davidson et al., 2012) also led to the increase in surface albedo,
thus, contributing −6.15% to the global ARF change between 1983 and 2010.
The differences (percentage change, %) in regional ARF between the two scenarios for
the nine regions are depicted by the brown dashed lines (scaled on the right Y-axis) in the line
charts of Fig. 2. Except for Europe, the Near East and North Africa, and South Asia, where the



annual ARFs are stronger than those derived from LUH1-LUC, the percentage changes in most regions exhibit the opposite phase of the ARFs predicted by OSCAR using GLASS-GLC. In East and Southeast Asia, for instance, the ARF derived from the S1 model scenario decreased from −0.028 W m$^{-2}$ in 1983 to −0.031 W m$^{-2}$ in 2010, indicating a reinforced cooling effect. In contrast, the ARF derived from the S2 scenario using the GLASS-GLC inventory exhibits the change from −0.027 W m$^{-2}$ in 1983 to −0.011 W m$^{-2}$ in 2010, indicating an attenuated cooling effect. The result suggests again that the LUH1-LUC inventory does not capture the change in LUC in East and Southeast Asia since the 1980s. Other details are presented in Table S5. Similar variations and trends of GLASS-GLC-driven ARFs can be observed in Russia, North America, and Oceania, where the negative ARFs exhibit rising trends from 1983 to 2010, indicating once again the declining negative ARF values and weakening cooling effect. In South Asia, the average percentage change in ARF between the two scenarios is the highest, at 37.30%. Moreover, Europe, the Near East and North Africa, and South Asia yielded positive ARFs. The increasing ARF trends indicate an intensification of the warming effect in these regions during this period. Sub-Saharan Africa experienced the greatest negative ARF values and fluctuations in both model scenarios. We found that the ARF from the S1 scenario extended from −0.057 W m$^{-2}$ in 1983 to −0.050 W m$^{-2}$ in 2010. The ARF from the S2 scenario dropped from −0.051 W m$^{-2}$ in 1983 to −0.061 W m$^{-2}$ in 2010. These results illustrate that the LUH1-LUC data attenuates the cooling effect in Sub-Saharan Africa, whereas the GLASS-GLC inventory enhances the cooling effect, demonstrating once again that the LUC data with significantly different resolutions and sources could alter the conclusions in the evaluation of LUC-induced climate forcing. Further details are provided in Supplementary Text 4 and Table S5.

**3.3. Effective Area of LU Conversion and Interannual ARF Variations**

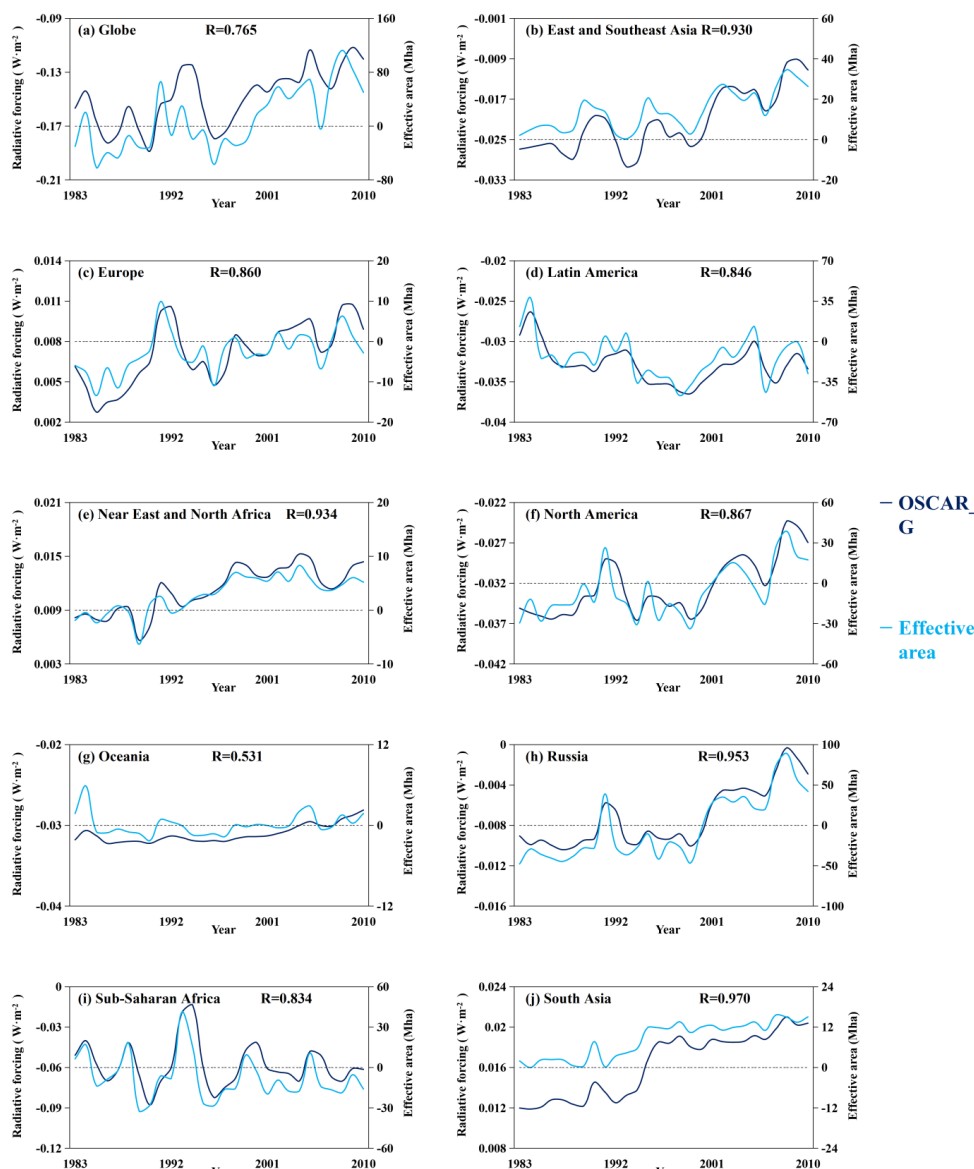

**Figure 3.** Annual RF (W m$^{-2}$) due to surface albedo change in model scenarios 2 from 1983 to 2010 derived from GLASS-GLC inventory (solid black line) and effective area (solid blue line) in the globe and nine



regions. (a) Globe; (b) East and Southeast Asia; (c) Europe; (d) Latin America; (e) Near East and North
Africa; (f) North America; (g) Oceania; (h) Russia; (i) Sub-Saharan Africa; (j) South Asia. Correlation
coefficients between the ARF and effective area are marked in the figures. The effective area measures the
extent of the area of all net LU conversion contributing to the change in ARF (see Methods).

We designed 20 sensitivity experiments to examine the contribution of LU conversion
among the five LU types to the variation in ARF from 1982 to 2010 (Methods, Table S1). We
introduced a disturbance capacity (DC, %), and an effective area (EA, see Eq. 4 in Methods)
to explain the changes in ARF caused by the size of LU conversion areas. Here, DC (%)
quantifies the extent of LU conversion that may considerably impact the change in ARF. The
EA is the sum of six net LU conversions that quantifies the extent of LU conversion
contributing to the change in ARF. In the sensitivity experiments, we reduced the area of LU
transition by 20% for each LU conversion (Gong et al., 2013). The model combines the ARF
results from 20 sensitivity experiments with the LUC for each target region. Further details are
provided in Methods and Supplementary Text 3. We also analyzed the rate and magnitude of
annual ARF fluctuations associated with EAs in the world and nine regions between 1983 and
2010. The details are presented in Supplementary Text 5 and Figs. S3–S12.
Figure 3 depicts the annual ARF (scaled on the left Y-axis) and EA (scaled on the right
Y-axis) in the globe and nine selected regions. As seen, the global annual ARF, which is the
sum of the ARFs in the nine regions based on OSCAR simulations, is stronger than regional
ARF due to the larger scale of global land cover change (Table S3 and Fig. S1) and stronger
albedo changes. The correlation coefficient between the ARF and EA in the globe is 0.765 (*P*-
value < 0.01), indicating that the net LU conversion area in the globe explains 59% of the
global ARF variation. In this instance, the global EA consists of a cumulative area of six net
LU conversion types. The percentage of individual LU conversions to the global EA is
presented in Table S4 and Fig. S13. As shown in Fig. S13a, the interannual fluctuations of the
EA (blue solid line) agree well with that of the transition area from the grassland-to-forest land
(red dashed line). Together with the grassland to cropland transition, these two LU transitions
contribute the most to the global LU transition, accounting for 52.5% of the total EA worldwide.



Such significant LU conversions have been attributed to grassland degradation (Bardgett et al.,
2021; Andrade et al., 2015; Aune et al., 2018; Berangere et al., 2018), such as the expansion
of croplands in the US, which reduced prairie grasslands (Lark et al., 2020). Since the surface
albedo of grassland is greater than that of forest and cropland (Edouard et al., 2010; Jackson et
al., 2008), grassland degradation could be considered a major contributor to the increase in
global ARF since the mid-1990s (Fig. 3a). This increasing ARF is crucial to the attenuation of
the cooling effect of the global negative ARF from −0.15 W m$^{-2}$ to −0.12 W m$^{-2}$. In East and
Southeast Asia, the correlation coefficient between ARF and EA is 0.930 ($P$-value < 0.01),
indicating that the EA explains 86% of the ARF change in this region. In particular, the
interannual fluctuation of the EA agree well with the LU transition from cropland to forestland
(green dashed line, Fig. S13b), followed by the transition from grassland to forestland (red
dashed line). Afforestation plays a key role in the ARF changes in East and Southeast Asia,
including grassland to forest, shrub to forest, and cropland to forest. These three LU transitions
account for 68.6% of total EA in this region. Previous reports have indicated that the forest
area in Southeast Asia has been decreasing in recent decades (Hansen et al., 2013; Achard et
al., 2002; Estoque et al., 2019) and has suffered from a net loss of 1.6 million ha yr$^{-1}$ (0.6%
yr$^{-1}$), thus, reducing the region's forest cover from 268 million ha in 1990 to 236 million ha in
2010 (Stibig et al., 2014). However, China has expanded the world's largest afforested area
since the late 1980s and had the world's largest artificial forest area in 2008, comprising
approximately 62 million hectares (National Forest Resource Inventory Report, 2009;
http://www.fao.org/forestry/fra/fra2010/en/, 2010; Zhang et al., 2015). Consequently, the
forest cover in East and Southeast Asia has expanded accordingly. Given the low albedo of
forested land (Igusky et al., 2008) and the forest land expansion over the past four decades
(Zhang et al., 2016; Peng et al., 2014), we observed a decreasing albedo and more positive
ARF in these regions. In Latin America, the correlation between ARF and EA is 0.846 ($P$-
value < 0.01). As shown in Fig. S13d, the annual variation of the EA nearly overlaps with the
transition zone between grassland and forest land. This LU transition contributes 77.4% to the
EA in Latin America, thereby playing a significant role in the ARF in this continent. In recent
decades, forest areas in Latin America have experienced a dramatic decline (Global Forest



Resources Assessment, 2020), partly due to forest wildfires (ARAGÃO et al., 2010; Escobar
et al., 2019) and the transition from forest lands to pastureland under the significantly rising
global demand for agricultural products (such as meat and soybeans) in this region.
Correspondingly, remarkable deforestation (Armenteras et al., 2019; Bullock et al., 2020) and
conversion of forest to grassland have been observed (Andela et al., 2017). Spanning almost
15 years (1990 to 2005), Latin America has been reported to have lost 7% of its forests (Da
Ponte et al., 2015). This transition resulted in an increase in albedo and a decrease in ARF in
Latin America (Fig. 3d).

The correlation coefficient between the ARF and the EA in Sub-Saharan Africa is 0.834

($P$-value < 0.01). The EA in this region consists of the cumulative area of five net LU
conversions (Table S4). The conversion between forestland and shrubs made the largest
contribution (48.9%) to the total EA. Sub-Saharan Africa is home to most of the world's
tropical grassy ecosystems (grasslands and savannas), comprising ~33.5% of Africa's landmass
(Parr et al., 2014). In recent years, the forest area in Sub-Saharan Africa has decreased
(Carherine et al., 2013), accompanied by an increase in savanna (including shrubs) (Atsri et al.,
2018; Gaillard et al., 2018). As depicted in Fig. 3i and Fig. S13i, declining forestland in Sub-
Saharan Africa consistently produces negative ARF, despite annual fluctuations.

In South Asia, the correlation coefficient between ARF and EA is 0.97 ($P$-value < 0.01),

with the EA including the cumulative area of six net LU conversions (Table S4). Of these LU
conversions, cropland-related LU transitions contributed up to 81.4% to the total EA. This
region of Asia has experienced the most successful Green Revolution since the late 1960s (Liu
et al., 2021), and India is one of the largest producers of agricultural commodities (FAOSTAT:
Food and agricultural data, 2017; Teluguntla et al., 2015), with more than half of its territory
used for cropland. Since the 1980s, the continuous expansion of cropland in South Asia (Hinz
et al., 2020) has led to a decrease in albedo, increasing ARF (Fig. 3j and Fig. S13j). Further
discussions on EA in Europe, the Near East and North Africa, North America, Oceania, and
Russia, as shown in Fig. S13, are presented in Supplementary Text 6.

**3.4 Response of Global ARF Change to Regional LUC Area and LU Conversion**

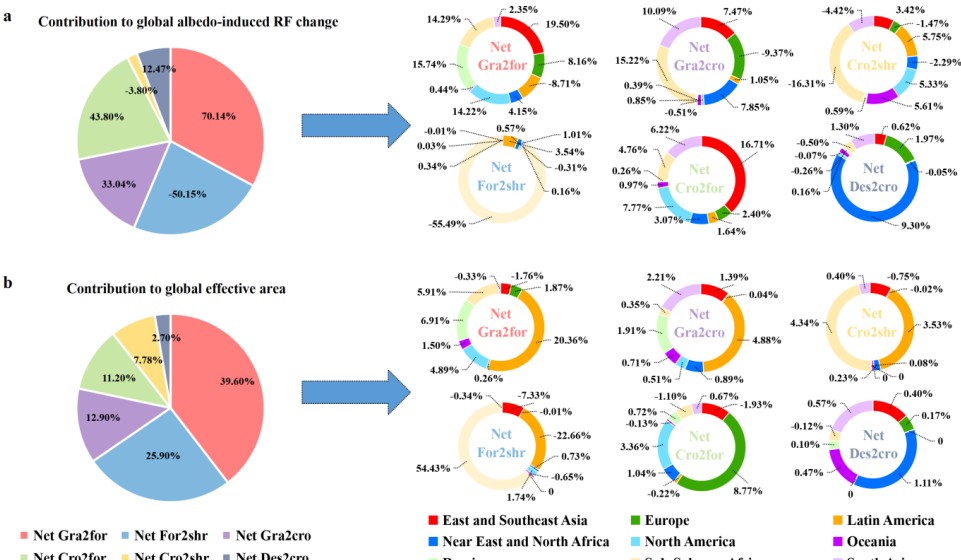

**Figure 4.** Contribution of six net LU conversion types in nine regions (|DC| > 1%) to the change in global ARF and EA globally and nine regions. (a) Pie charts on the left panel show the contribution of six LU conversion types in nine regions to the change in global ARF, including grassland to the forest (orange), forest to shrubs (light blue), grassland to cropland (light purple), cropland to the forest (light green), cropland to shrubs (light yellow), and desert to cropland (light gray). Donut charts on the right panel show the contribution of each of the six net LU conversion types in each of the nine regions to the change in global ARF. Among the nine regions, East and Southeast Asia are colored red, Europe deep green, Latin America deep yellow, Near East and North Africa deep blue, North America blue, Oceania purple, Russia green, Sub-Saharan Africa yellow, and South Asia purple gray. The coefficient of variation (CV) is ±5%. (b) Contribution of six LU conversions (|DC| > 1%) in the nine regions to global EA.

To quantify the contribution of each LU transition in each region to the changes in global ARF and EA, we estimated ARF and EA changes without LU conversion from 1983 to 2010 by reducing LU transition areas from 20% to 100% in 20 sensitivity experiments (see Methods), indicating no LU transition. Subsequently, we calculated the differences between ARF and EA changes with and without LU conversion to determine the contributions of any LU conversion in any region to the changes in global ARF and EA, as defined by $C_{ARF}$ and $C_{EA}$, as described



in Eqs.5–9 of Methods. The net conversion of grasslands to forests contributed 70.14% to the
change in the global ARF from 1983 to 2010. During this period, the global ARF increased by
0.036 W m$^{-2}$, in line with the general upward trend. Since the albedo of grasslands is greater
than that of forests, we would anticipate a decrease in albedo during the transition from
grasslands to forests, which tends to increase the ARF. Efforts have been made to increase the
global forest cover through afforestation programs. However, most afforestation programs
have been implemented at the expense of natural vegetation, particularly grasslands, rather than
agricultural land (Berangere et al., 2018; Zablon et al., 2018). Globally expansive grasslands
were found to be suitable for future forest restoration programs to offset anthropogenic $CO_2$
emissions (Bond et al., 2016). With the updated LUC inventory with the satellite measured
information on a fine temporal-spatial scale, we could assess the effect of increasing forest
coverage on ARF with greater precision. The donut charts on the right side of Fig. 4a depict
the change in global ARF due to LU conversions in each of the nine regions. The results
indicate that grassland to forest conversion in East and Southeast Asia contributes 19.50% to
the change in global ARF, 8.16% from Europe, −8.71% from Latin America, 4.15% from the
Near East and North Africa, 14.22% from North America, 0.44% from Oceania, 15.74% from
Russia, 14.29% from Sub-Saharan Africa, and 2.35% from South Asia, respectively. The
global net conversion of forests to shrubs contributes −50.15% to the change in global ARF,
with individual contributions from East and Southeast Asia (0.16%), Europe (−0.01%), Latin
America (3.54%), Near East and North Africa (1.01%), North America (0.57%), Oceania
(−0.31%), Russia (0.03%), Sub-Saharan Africa (−55.49%), and South Asia (0.35%). Thus, the
contribution of the net conversion of forests to shrubs to the global ARF change enhanced the
cooling effect. The contributions from the remaining four net conversion types are shown in
Table S6.

The contributions of net conversion from grassland to forest, forest to shrubs, grassland

to cropland, cropland to the forest, cropland to shrubs, and desert to cropland to the global EA
(Eq. (4)) are 39.60%, 25.90%, 12.90%, 11.20%, 7.78%, and 2.70%, respectively, as depicted
in the pie charts on the left panel of Fig. 4b. The contributions of the six net conversion types
in each of nine regions to the global EA are displayed in donut charts in the right panel of Fig.





4b, providing additional information regarding the impact of regional LU conversion on the
variation in global ARF. For instance, the contributions of net conversion type of grassland to
forest in each of the nine regions to the global EA are as follows: −1.76% from East and
Southeast Asia, 1.87% from Europe, 20.36% from Latin America, 0.26% from Near East and
North Africa, 4.89% from North America, 1.49% from Oceania, 6.91% from Russia, 5.91%
from Sub-Saharan Africa, and −0.33% from South Asia. Additional results for the remaining
five types of net conversion in each of the nine regions are presented in Table S7.
**3.5 Contributions of Two-way LU Conversion to Global ARF Change**

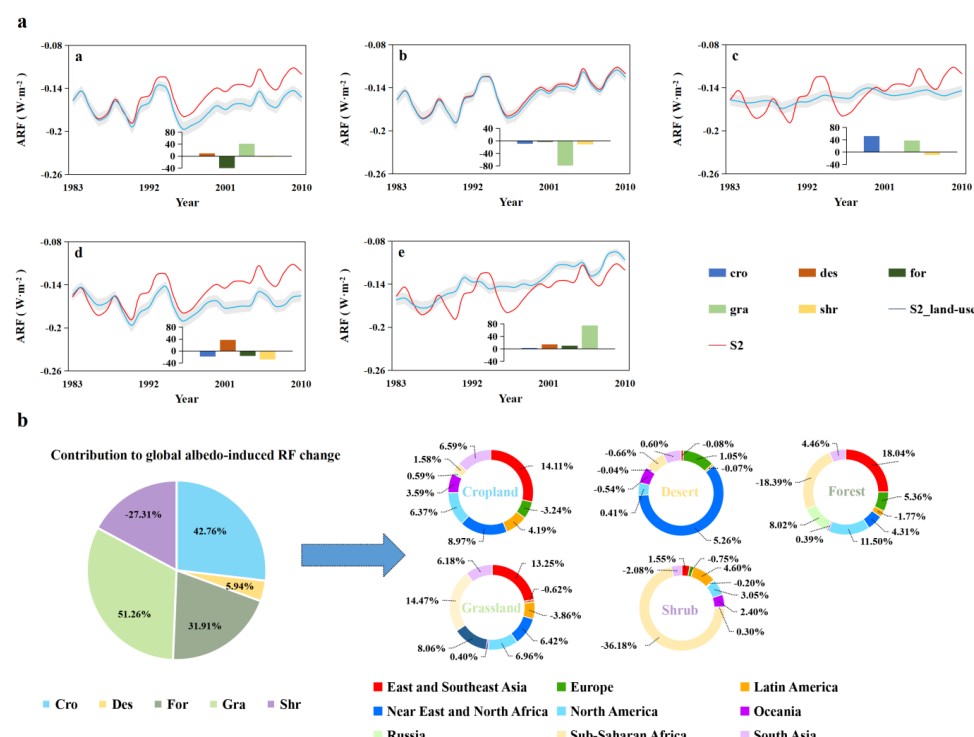


**Figure 5.** Changes in global ARF derived from model scenario 2 (S2) and contribution of five LU types in
the globe and nine regions to the change in global ARF. (a) Changes in global ARF subject to LU transition
from 1983 to 2010 (solid red line) and a fixed LU type without transition (solid blue line) for five LU types,





including croplands (cro, Fig. 5a-a), deserts (des, Fig. 5a-b), forests (for, Fig. 5a-c), grasslands (gra, Fig. 5a-
d), and shrublands (shr, Fig. 5a-e). The inset bar chart represents the relative contribution of the two-way
LU net transition between the LU of the interested and other LUs from 1983 to 2010. Taking the bar chart
in Fig. 5a-a as an example, the bars with different colors show the result of the two-way transition between
cropland and other LU types from 1983 to 2010. Positive bars represent the conversion from other LU types
to cropland, and negative bars indicate the transition from cropland to other LU types. Shadings in Fig. 5a-
a–5a-e indicate the uncertainty interval estimated by Monte Carlo simulations. (b) Contribution of five LU
types in each region to changes in global ARF. The pie chart on the left panel shows the contribution of five
LU types to the change in ARF in the globe, including cropland (light blue), desert (light yellow), forest
(gray-green), grassland (light green), and shrubs (light purple). Five small donut charts on the right panel
show the contribution of each type in each of the nine regions to changes in global ARF. The coefficient of
variation (CV) is ±5%.

We also set up 20 sensitivity experiments to examine the response of ARF to two-way LU

transition in each region. The two-way LU transition entails LU conversion from a particular
LU type to the remaining 4 LU types and vice versa, which accounts for eight LU conversions
for the five LU types in the GLASS-GLS inventory. We compare the changes in global ARF
driven by LU transition from 1983 to 2010 to the ARF estimated by reducing LU transition
areas from 20% to 100%. The 100% reduction of LU transition area means no LU transition.
As an illustration, Fig. 5a-a compares the change in global ARF caused by the transition
between cropland and the other four LUs (cropland to desert, forest, grass, and shrub, solid red
line) and without transition (fixed cropland, solid blue line). Marked differences can be
observed for both with and without the transition between croplands and other LU types. The
trend and annual fluctuation of ARF are consistent with the results subject to LU transition
(solid red line). However, under the fixed cropland (no LU transition) during this period (solid
blue line), the negative values of ARF change have decreased since 1990 in comparison to the
case with LU transition. As shown in the inset of Fig. 5a-a, the transition from grassland to
cropland accounts for 41.0% of the cropland transition area, while the net transition from desert
to cropland accounts for 9.2%. The remaining two net LU transitions occurred from croplands
to forests (−48.9%) and shrubs (−0.9%), respectively, implying that LU transitions from
croplands to other LUs account for −49.8% of the cropland transition area. By combining these



transition areas, the net cropland transition area was calculated to be 0.4%, indicating the growth of cropland (Fig. S1). Since the transition from grasslands to croplands decreased surface albedo (Table S2), the LU conversion in this instance decreases the absolute value of negative ARF, thereby weakening the cooling effect.

Using the GLASS-GLC inventory (scenario 2), we further estimated the percentage change (%) in global ARF with the transition between cropland and the other four LUs (solid red line, Fig. S14a) from 1983 to 2010. During this period, the percentage changes ranged from −0.6% to 28.4%, illustrating a significant upward trend. From 1998 to 2010, the annual percentage change in global ARF was almost 15%, indicating that the cropland transition significantly contributed to the change in global ARF. We also observed an overall increase in cropland area from 1983 to 2010, as indicated by the positive accumulated cropland area in Fig. S14a (solid black line, scaled to the right of the Y-axis), which is consistent with the growth rate of cropland area of 0.037 Mha/yr during this period.

Similarly, Fig. 5a-b–5a-e illustrate OSCAR-modeled global ARF variation utilizing GLASS-GLC inventory with and without LU transition of individual LU types from 1983 to 2010. As shown in Fig. 5a-b, the conversion of the desert to other LU types has little effect on the global ARF variation, and the modeled ARF from simulations with and without LU transition is nearly identical. Fig. S14b demonstrates that the percentage change in the global ARF was less than 4.2% between 1983 and 2010, with a mean value of 1.7%. As illustrated in the inset of Fig. 5a-b, the net transition from desert to grassland (light green bar) accounts for 79.1% of the total transition area, the net transition from desert to shrubs (light yellow bar) accounts for 11.5% of the total transition area, and the net transition from desert to cropland (deep blue bar) accounts for 8.9% of the total transition area, respectively. The percentage change indicates a net decrease in a desert land, which is supported by the declining accumulated area of desert land (Fig. S14b). The lack of significant differences between global ARF with and without LU transition is most likely due to the smaller change in the desert area over the past decades. Detailed discussions of the variations in ARF induced by forest, grassland, and shrub transitions utilizing the GLASS-GLC inventory, as depicted in Fig. 5a-c– 5a-e are presented in Supplementary Text 7. Overall, Fig. 5a reveals an increasing trend of





ARF change, highlighted by attenuated negative ARF from 1983 to 2010, which suggests a
weakening cooling effect by the global ARF (Fig. 1).
Fig. 5b depicts the contribution of regional LU transactions to the change in global ARF
(five pie charts on the right panel of Fig. 5b). Cropland, desert, forest, grassland, and shrubs
contributed 42.76%, 5.94%, 31.91%, 51.26%, and −27.31%, respectively, to the change in
global ARF, as depicted in the pie chart on the left panel of in Fig. 5b. The donut charts on the
right panel of Fig. 5b illustrate the contribution of each of the five LU types in the nine regions
to the change in the global ARF. Taking cropland as an example, the contributions of cropland-
related conversions in each of the nine regions to the change in global ARF are as follows:
14.11% (East and Southeast Asia), −3.24% (Europe), 4.19% (Latin America), 8.97% (Near
East and North Africa), 6.37% (North America), 3.59% (Oceania), 0.59% (Russia), 1.58%
(Sub-Saharan Africa), and 6.59% (South Asia). As stated previously and depicted in the left
panel of Fig. 5b, the sum of the contributions from these nine regions to the global ARF change
is 42.76%. As a result, cropland-related LU conversion in East and Southeast Asia (primarily
China) made the largest contribution to global ARF variation. The results for the remaining
four LU types are presented in Table S8.

**4. Discussion**
By incorporating a recently developed satellite-remote sensing-based high-resolution
LUC dataset into the OSCAR model, we demonstrate that previous estimates of ARF derived
from historical statistics-based LUH1-LUC data with a coarse resolution tend to overestimate
the LUC driving albedo-induced cooling effect. Our revised estimate reveals that the global
ARF ($-0.12$ W m$^{-2}$) is lower than the value adopted by the IPCC ($-0.15$ W m$^{-2}$). Our results
indicate that, among the nine selected regions covering the global land area, Sub-Saharan
Africa made the largest net contribution (39.2%) to the global mean ARF (-0.06 W m$^{-2}$) owing
to the transition of forestland to shrubland, which result in greater surface albedo and, hence,
declining ARF. The latter became very significant from 1982 to 2010. East and Southeast Asia
also contributed significantly, following Sub-Saharan Africa, to the changes in global ARF at



33.6% (0.016 W m$^{-2}$) due to the LU conversion from the grassland to forest conversion and
land desertification management, which result in lower surface albedo (Table S2) and
increasing ARF. In line with previous researches, we demonstrate that RF induced by changes
in surface albedo is primarily driven by changes in vegetation (Betts et al., 2000). The
transformation from forest to grass, shrub, and crop, and crop to grass resulted in decrease in
ARF of −0.68 W m$^{-2}$, −0.48 W m$^{-2}$, –0.19 W m$^{-2}$, and –0.22 W m$^{-2}$, respectively, due to the
enhancement of surface albedo by the transformation from forest to these vegetation types.
Opposite conversions of these vegetation types to forests outweigh positive contributions to
ARF, indicating a rise in surface albedos and cooling effects. In addition to the magnitudes, we
find that the two LUC datasets developed based on different data sources, approaches, and
resolutions produce different ARFs, indicating that LUC data influenced considerably on
regional and continental ARFs.
Notably, the present study only predicts ARF and its change induced by surface albedo
subject to LUC and LU conversions but does not address RF driven by $CO_2$ emissions as a
result of carbon source-sink conversions associated with LUC and the ARF associated with the
LULCC-induced changes in snow cover. However, the major findings of dominant LU
transition patterns between forest and grassland/shrub/cropland imply $CO_2$ source-sink
transitions, which are expected to influence LUC-driven RF more strongly. On the one hand,
the unexpectedly weaker cooling effect of LUC observed in this study indicates that global LU
and LU conversion as carbon sinks since the 1980s do not significantly mitigate climate
warming. On the other hand, land management must be improved by increasing the capacity
of LUC for carbon sequestration, preserving carbon sinks, and providing renewable resources.
Our results show that Sub-Saharan Africa contributed the most to the forest-to-grass and forest-
to-shrub transition-induced global ARF, with predicted ARF values of −0.20 W m$^{-2}$ and −0.40
W m$^{-2}$, respectively. In addition, East and Southeast Asia contribute the most to the ARF due
to the conversion of LU from forest to crop and crop to grass. Furthermore, Sub-Saharan Africa
has also been confirmed to have the highest proportion of forest-to-grass and forest-to-shrub
transitions, contributing to a cooling effect.





These findings have substantial ramifications for pertinent policy issues. Accordingly, they suggest that local governments and international communities should take more action in Sub-Saharan Africa to slow down or, preferably, stop deforestation and forest-to-grassland-and-cropland conversion, which is a significant contributor to carbon emission enhancement (Spawn et al., 2019; Pendrill et al., 2019; Chang et al., 2021). In our case, even though this LU transition increases surface albedo, thereby increasing LUC-albedo–induced negative RF and exerting a cooling effect, this effect is negligible compared to the increase in RF caused by $CO_2$ emissions (IPCC AR6, 2021; Li et al., 2016; Jian et al., 2022). Therefore, the cooling effect of afforestation on reducing $CO_2$ emissions outweighs the warming effect of the resultant decrease in surface albedo. The crop-to-forest transition occurring primarily in East and Southeast Asia, Europe, and the Near East and North Africa has been partially encouraged by national and international cropland and water resource conservation strategies and programs, resulting in ARF values of 0.09 W m$^{-2}$, 0.02 W m$^{-2}$, and 0.01 W m$^{-2}$, respectively. The "Grain-for-Green" program in northwestern China (Wang et al., 2023), for example, impedes the transition from crop to forest in East and Southeast Asia. Although the program helps improve the ecological environment, from the perspective of ARF, it tends to reduce the surface albedo and increase positive RF, thereby enhancing the warming effect. It is worth noting that the present study did not incorporate non-radiative process and the coupling between land and atmosphere, which might drive many feedback mechanisms. The significance of land management in maintaining carbon sinks and providing renewable resources was also not dealt with. However, this study provides additional evidence of the importance of land management in influencing the carbon sinks. Optimal land management should implement integrated and enforceable sustainable agriculture, climate-smart forestry, and climate-friendly land resources with co-benefits and cost-efficiency.

## 5. Conclusions

We have improved the global and the nine regional ARF simulations using OSCAR model a updated LUC dataset on a high temporal-spatial resolution. We explored the causes of ARF changes in the world and nine regions across the globe by disentangling land change data for





20 transformation types. We also developed the concepts of DC and EA to better explain the
changes in ARFs. The major findings are summarized below:
• The magnitude of the negative ARF obtained from this study is 20% lower than previous
estimations, implying a weaker cooling effect. The results suggest that global LUC-
induced changes in surface albedo may not significantly slow global warming as
previously expected.
• Sub-Saharan Africa made the largest net contribution to the global ARF (39.2%) due to
significant land-use conversions, typically from forest to other vegetation land
accompanying with higher surface albedo. The most significant land cover changes
occurred in East and Southeast Asia, which dominated (33.6%) the changes in global ARF
in recent decades.
• The largest change in global ARF occurs in the net transition from grassland to forest,
contributing 70.14% to LUC-induced ARF. Of which, East and Southeast Asia region
accounts for 19.50% of the change in global ARF. The net transition from forest to shrub
made the largest negative contribution of -50.15% to the LUC-induced change in global
ARF, of which Sub-Saharan Africa accounted for -55.49% to the change in global ARF.
• Vegetation lands exert a most vital effect on global ARF variation, of which grassland
contributed 51.26%. Among those vegetation lands, the changes in grasslands in Sub-
Saharan Africa contributed 14.47% to the global ARF variation subject to the vegetation
land transition, followed by East and Southeast Asia at 13.25%.
**Code availability**
OSCAR v2.4 source code is available for downloading on https://github.com/tgasser/OSCAR.
**Data availability**
GLASS-GLC data can be accessed at https://doi.org/10.1594/PANGAEA.913496.
**Author contributions**
All authors contributed to the manuscript and have given approval of the final version. XZ



coordinated and supervised the project. XZ, XJ and JM designed the present experiment, carried out modeling, and drafted the manuscript. HG, YZ and RZ collected the data. XL, KC, TH, ST and JL analyzed simulation results.

**Competing interests**

The authors declare that they have no known competing financial interests or personal relationships that could have appeared to influence the work reported in this paper.

**Acknowledgements**

We wish to thank the High-performance Computing Platform of Peking University to support extensive model simulations of this study. We acknowledge the use of OSCAR model and GLASS-GLC dataset.

**Financial support**

This study is supported by National Key R&D Program of China (2023YFE0112900) and the National Natural Science Foundation of China (42407134, 41991312, and 41977357).

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
