# Peer review of "Highly-resolved satellite remote sensing based land-use change inventory"

_EGUsphere, 2024_

## Author Comment (AC1)

**Responses to Editor and Referee's comments**

First of all, we would like to thank the Editor and Referee for their comments and suggestions, which improved greatly the presentations and interpretations in our revised manuscript. In the revised article, we have addressed all comments and suggestions from the Editor and Referee. Our point-by-point responses to the Referee's comments are outlined below. The Referee's original comments are shown in italics and our responses are given in normal fonts.

**Referee #3**

**Comments:**
*This manuscript, which estimates albedo-induced radiative forcing (ARF) using satellite-derived land-use change (LUC) data at fine spatial resolution, has the potential to significantly impact our understanding of LUC effects. The authors' report of a lower ARF estimation using fine spatial resolution data than published values, suggesting a weaker cooling effect of LUC, is a promising finding. The manuscript is well written and interesting, the motivation to the work is strong, the methodology is well described, and the figures are engaging and effective.*

**Response:** We thank the Referee's positive and encouraging comments which help us to improve this article considerably.

**Main concerns:**

*1. As the authors have mentioned, LUH2 is more recent and at a finer spatial resolution than LUH1. Despite this, why was LUH2 data not used instead of LUH1 for comparison to GLASS-GLC?*

**Response:** The LUH2 dataset collectively categorizes grass, shrub, and other surface types into Non-forest types, and the purpose of this study is demonstrate that highly-resolved LUC data could yield significant different RF from previous investigations, and to comprehensively assess the effects of transformation among major LU types. Molded RFs induced by land-use changes using LUH2 and GLASS-GLC might not be consistent due to different LU categories. In fact, the LUC-derived global RF reported in IPCC AR6 used LUH2 dataset (Fig. 1b).

*2. Temporal variation in LUC appears large for all regions, resulting in large fluctuations in ARF (Figure 1 and 2). Such large variations in LUC should be justified or studies reporting similar fluctuations should be cited.*

**Response:** Following Reviewer's comment, we compared GLASS-GLC and MODIS LUC data, of which, the GLASS-GLC used satellite multi-source fusion approach and MODIS used direct MODIS sensor to derived their respective LUC inventories. The GLASS-GLC dataset spanning 1982-2015 but MODIS data is only available from 2000

onward. So, we replaced the GLASS-GLC by MODIS LULC data from 2002 to 2010 in the OSCAR model. The figure below shows annual fluctuations of the OSCAR simulated annual RF under global forestland changes using GLASS-GLS and MODIS from 2002 to 2010, respectively. Both RF results show annual fluctuations, though the RFs from the CLASS-GLC illustrate somewhat stronger oscillations. However, during this period, accumulated RFs subject to the global forestland changes driven by GLASS-GLC and MODIS LUC are 0.0165 $Wm^{-2}$ and 0.0157 $W\ m^{-2}$, respectively, indicating only a 5% difference between the two satellite remote sensing derived LUC datasets.

Sun et al. (2022) compared the applications of six LULC products in the identification of LUCs in Northwestern China. Their results indicated, while the GLASS-GLC and MODIS (MCD-12Q1) were not superior to other four products (developed only for China), these two datasets were of most temporal and spatial consistency. This paper has been cited in the revised paper.

These discussions have been summarized in a new paragraph in section 2.2 (third paragraph).

[Figure]

Sun, W. et al. Land use and cover changes on the Loess Plateau: A comparison of six global or national land use and cover datasets. Land Use Policy 119, 106165 (2022).

**Additional comment:**

*The authors have provided websites for downloading the GLASS-GLC data and OSCAR code but have not shared a repository to access the outputs of OSCAR model generated and analyzed in this study. I would encourage them to share a link to their model simulations.*

**Response:** Done, thanks! We provide a repository to access relevant output data (https://doi.org/10.5281/zenodo.14586249).

---

## Author Comment (AC2)

**Responses to Editor and Referee's comments**

First of all, we would like to thank the Editor and Referee for their comments and suggestions, which improved greatly the presentations and interpretations in our revised manuscript. In the revised article, we have addressed all comments and suggestions from the Editor and Referee. Our point-by-point responses to the Referee's comments are outlined below. The Referee's original comments are shown in italics and our responses are given in normal fonts.

**Referee #2**

**Comments:**
*The manuscript egusphere-2024-1497 introduces a satellite-derived historical land cover product to a climate model, recalculates the radiative forcing (RF) of land use change (LUC) from 1983 to 2010, and demonstrates that satellite-derived results show weaker LUC RF compared to the original model's coarse-resolution LUC input. This study is well designed, and the results sufficiently support the conclusions. I have some questions regarding the interpretation of the results, and I believe that addressing these concerns will strengthen the manuscript and facilitate its publication in ACP.*

**Response:** We thank the Referee's positive and encouraging comments, which help us to improve this article considerably.

**Major Comments**

*1. Inter-Annual Variability of Satellite Land Cover Product*
*My main concern is the excessive annual variability in the satellite-derived land cover product, particularly when it is claimed to represent land use change. Typically, land use change reflects human activities. However, Figure 1 shows significant fluctuations in the global average of LUC-derived RF between the late 1980s and early 1990s, with increases and decreases that nearly double the overall magnitude observed since the industrial era. Similar abrupt changes are noted in South Asia and Russia in the late 1990s (Figure 2). These fluctuations seem unrealistic and undermine the reliability of the input satellite data. I recommend exploring additional satellite datasets, if available, and comparing the results for inter-validation.*

**Response:** Following Reviewer's comment, we compared GLASS-GLC and MODIS LUC data, of which, the GLASS-GLC used satellite multi-source fusion approach and MODIS used direct MODIS sensor to derived their respective LUC inventories. The GLASS-GLC dataset spanning 1982-2015 but MODIS data is only available from 2000 onward. So, we replaced the GLASS-GLC by MODIS LULC data from 2002 to 2010 in the OSCAR model. The figure below shows annual fluctuations of the OSCAR simulated annual RF under global forestland changes using GLASS-GLS and MODIS from 2002 to 2010, respectively. Both RF results show annual fluctuations, though the

RFs from the CLASS-GLC illustrate somewhat stronger oscillations. However, during this period, accumulated RFs subject to the global forestland changes driven by GLASS-GLC and MODIS LUC are 0.0165 Wm$^{-2}$ and 0.0157 W m$^{-2}$, respectively, indicating only a 5% difference between the two satellite remote sensing derived LUC datasets.

Sun et al. (2022) compared the applications of six LULC products in the identification of LUCs in Northwestern China. Their results indicated, while the GLASS-GLC and MODIS (MCD-12Q1) were not superior to other four products (developed only for China), these two datasets were of most temporal and spatial consistency. This paper has been cited in the revised paper.

These discussions have been summarized in a new paragraph in section 2.2 (third paragraph).

[Figure]

The GLASS-GLC dataset was further compared temporally and spatially with the LUH1 dataset in Supplementary Figures S1 and S2. While the GLASS-GLC is superior to the LUH1, the magnitude of GLASS-GLC is comparable to LUH1 dataset. Eq. S1 defines the principle of the OSCAR model to predict ARF, which is closely related to the area of LUC, and therefore, the fluctuation of the ARF results is also reflected by the land use conversion data of the dataset, which is well reflected by Figure 3 and Figs. S3-S13 of this paper.

The result has been added to the revised Supplementary Text 2 (the last paragraph).

Sun, W. et al. Land use and cover changes on the Loess Plateau: A comparison of six global or national land use and cover datasets. Land Use Policy 119, 106165 (2022).

2. *Land Cover and Land Use Classification*
*How do the authors reconcile the differences between the satellite-derived land cover classifications and the land use classifications in the original model input (LUH1)? Land cover and land use are distinct concepts, and their categories differ. For example,*

*LUH1 includes "pasture" as a category, while GLASS-GLC uses "grassland," which are not equivalent. Clarification on the mapping or harmonization process is needed.*

**Response:** The Reviewer raised an important question! Land Cover refers to the physical and biological cover over the surface of the Earth, including vegetation, water bodies, urban areas, and bare soil. For example, land cover categories include forest, grassland, water, built-up areas, and bare soil. Land Use refers to how land is used by humans, including agricultural practices, urban development, forestry activities, and conservation. Land use categories may include crop land, pasture, urban, and nature reserves.

The key differences partly come from terminological differences, namely, different datasets might use different terminologies for similar land cover types (e.g., "pasture" in LUH1 vs. "grassland" in GLASS-GLC). These terms may have specific implications in the context of land use and ecology. Dynamically, land use might change more rapidly due to socio-economic factors than land cover, leading to discrepancies between the two categories over time.

In the revised paper, we have inserted a new Table S1 in Supplementary and referred it in the second paragraph in section 2.2.

*3. Sensitivity Analysis Methodology*
*The sensitivity analysis is a critical foundation for this study. Is the method employed here commonly used for quantifying LUC radiative forcing? If not, how does it compare with approaches used in previous studies? Providing context and justification for this methodology is essential.*

**Response:** The reviewer's comment raises a good point about the sensitivity analysis method. We used the normalized marginal attribution method in the sensitivity analysis (Supplementary Text 8). This approach has been applied previously to assess sensitivity of OSCAR simulated radiative forcing (Li et al., 2016; Fu et al., 2020). Table S5 provides detailed analysis results, which are referred to in the end of section 3.5 and revised section 2.3.

1. Li, B. G. et l. The contribution of China's emissions to global climate forcing. *Nature* **531**, 357–361 (2016).
2. Fu, B. et al. Short-lived climate forcers have long-term climate impacts via the carbon–climate feedback. *Nat. Clim. Chang.* **10**, 851–855 (2020).

**Other Comments**

*1. Abstract: Clarify the apparent contradiction between "Sub-Saharan Africa made the largest net contribution" and "East and Southeast Asia dominated the changes in global ARF."*

**Response:** Here, the former refers to sub-Saharan Africa, which has the largest proportion of ARF to the value of global ARF, and the latter refers to East and Southeast Asia, which has the largest contribution to the change in global ARF, as showed in Figure 2. We have rewritten "*contribution*" as "*proportion*".

*2. Lines 55–57: Elaborate on the distinction between the well-investigated "LUC on climate balance" and the research gaps in "LUC-induced climate forcing."*

**Response:** `LUC on climate balance` refers to understanding how changes in land use (such as deforestation, urbanization, or land restoration) affect the climate system's energy balance through biogeochemical and biogeophysical processes. These processes drive **carbon sequestration and emissions and surface albedo change and interactively affects climate balance.** The `LUC-induced climate forcing` focuses on understanding how changes in land use generate direct climate forcing effects, which is an area that still requires significant exploration and has several research gaps, including mainly the lack of the long-term effects of LUC-induced climate forcing prediction. Immediate impacts may be different from those observed over decades or centuries.

Corresponding text has been added to the first and second paragraphs in revised Introduction.

*3. Lines 74–75: Suggest investigating multiple satellite products, rather than relying on a single dataset.*

**Response:** Please refer to our response to the Reviewer's major comment 1 and discussions in revised Supplementary Text 2 (last paragraph).

*4. Section 2.1: Provide an introduction to how OSCAR converts land use types into albedo values and their subsequent effects on radiative forcing and climate.*

**Response:** Thanks to Reviewer 1 for the suggestion. We provide a thorough description of how OSCAR converts land-use types into albedos in the simulation of the ARF. We have added a new (2nd) paragraph in Supplementary Text 1, which provides a detailed introduction to the issue raised by the Reviewer.

*5. Line 188: Explain the rationale for using a 20% threshold in the analysis.*

**Response:** For many satellite-derived land-use classification products, overall classification accuracies range between 70% and 90%. This implies misclassifications can lead to an uncertainty of 10% to 30% in land-use area estimates. So, we took 20% in our sensitivity experiments.

Corresponding text has been added to the first paragraph of revised section 2.3. We also added a reference (Gong et al., 2013) to the corresponding text.

Gong, P. et al. Finer resolution observation and monitoring of global land cover: first

mapping results with Landsat TM and ETM+ data. *Int. J. Remote Sens*. **34**, 2607–2654 (2013).

*6. Figure 1(b): Indicate the time periods covered by other studies for better comparability.*

**Response:** Done, thanks!

*7. Figure 2: Justify the chosen regional separations and clarify whether latitude weighting was applied to calculate the regional means.*

**Response:** The OSCAR model already separated the world into 114 countries and regions and this study further divided the globe into nine regions, each includes certain number of countries and regions (Table S3). The mean ARF value in each of the nine regions was obtained by averaging ARFs over those countries and regions grouped in each of the nine regions, not from latitude weighting. The corresponding text have been added to the first paragraph of section 3.2.

*8. Lines 347–349: This statement is unnecessary and could be removed to streamline the manuscript.*

**Response:** Done, thanks!

*9. Supplementary Table S2: Explain why the values for "Rest of East Asia" are notably larger than those for other regions.*

**Response:** Firstly, the parameters are provided by the OSCAR model, and secondly, the "*Rest of East Asia*" mainly indicates Mongolia, and the relevant albedo data also indicate that the surface albedo is high in this region (https://www.geodata.cn/).

---

## Author Comment (AC3)

**Responses to Editor and Referee's comments**

First of all, we would like to thank the Editor and Referee for their comments and suggestions, which improved greatly the presentations and interpretations in our revised manuscript. In the revised article, we have addressed all comments and suggestions from the Editor and Referee. Our point-by-point responses to the Referee's comments are outlined below. The Referee's original comments are shown in italics and our responses are given in normal fonts.

**Referee #1**

**Comments:**

*Land use change has been demonstrated to show large impacts on regional or even global climate change. This study quantified the LUC-induced albedo change and its radiative forcing based on high-resolution remote sensing-derived LUC dataset. Thank the authors carefully resolved my comments in the previous round of review. Please see below for my further comments.*

**Response:** We thank the Referee's positive and encouraging comments which help us to improve this article considerably.

**Major concerns:**

*1. Line 102-103: The authors stated that they assigned a 5% uncertainty in OSCAR modeled ARF based on LUC data uncertainty. However, what is the LUC data uncertainty? How did the authors derive this 5% threshold? Please provide more details.*

**Response:** The uncertainty of the LUC data is subject to its accuracy (82.81%, Liu et al., 2020). We examined the response of modeled ARF to the 5% uncertainty by increasing the uncertainty to 10% and 15%. The differences of simulated mean ARF between 5% and 10% and 15% were only 0.23% and 0.47%, respectively.

Corresponding text has been added to revised section 2.1.

Liu, H. et al. Annual dynamics of global land cover and its long-term changes from 1982 to 2015. *Earth Syst. Sci. Data* **12**, 1217–1243 (2020).

*2. Line 152: The authors mentioned that they conducted extensive sensitivity experiments by reducing each LU transition area by 20% within five major LU types. However, why did the authors select this 20% threshold? Please clearly clarify it.*

**Response:** For many satellite-derived land-use classification products, overall classification accuracies range between 70% and 90%. This implies that misclassifications can lead to an uncertainty of 10% to 30% in land-use area estimates.

So, we took 20% in our sensitivity experiments.

We have added corresponding text and a reference (Gong et al., 2013) in the first paragraph of section 2.3.

Gong, P. et al. Finer resolution observation and monitoring of global land cover: first mapping results with Landsat TM and ETM+ data. *Int. J. Remote Sens.* **34**, 2607–2654 (2013).

*3. Line 142: The authors mentioned that they neglected the LUC-induced surface roughness change. Please discuss the potential uncertainty from this.*

**Response:** Following the Reviewer's comment, we have rephrased the last paragraph in revised section 2.2, we wrote:

"The OSCAR model does not take the surface roughness length into account. The surface roughness affects primarily on turbulent exchange of heat and air mass between the underlying surface and air, which may indirectly alter surface radiation fluxes via changing sensible and latent fluxes under a heat balance status (Andrews, 2012). This characteristic can significantly influence RF largely via its association with surface albedo. Given that the OSCAR introduces directly the surface albedo, it is expected that excluding the roughness length would not perturbate RF prediction significantly."

*4. The latest LUH2 dataset is available. There are some improvements in LUH2 compared to LUH1. Please use the latest version of LUH2 rather than the out-of-date LUH1.*

**Response:** We thank the Reviewer's suggestion! The LUH2 dataset collectively categorizes grass, shrub, and other surface types into Non-forest types, and the purpose of this study is demonstrate that highly-resolved LUC data might yield significant difference of RF from previous investigations, and to comprehensively assess the effects of transformation among the major LU types on RF. Molded RFs induced by land-use changes using LUH2 and GLASS-GLC might not be consistent and, therefore, LUH2 results are not straightforward to compare with GLASS-GLC derived RF. In fact, the LUC-derived global RF reported in IPCC AR6 used LUH2 dataset (Fig. 1b).

*5. Although the GLASS-LUC has a higher spatial resolution, the authors upscaled them to national and regional levels. Please clearly clarify this point.*

**Response:** Because the OSCAR is not a grid-resolved model, the highly-resolved LUC data cannot be implemented directly into the model. However, the highly-resolved GLASS-GLC data provides more detailed LU type transition in each country, which plays a crucial role in estimating albedo-induced RF.

*6. However, GLASS-LUC also include uncertainties, and is not necessarily more accurate than LUH2 data. I suggest the authors include more remote sensing datasets e.g., MODIS data to increase the robustness of the results.*

**Response:** Following Reviewer's comment, we compared GLASS-GLC and MODIS LUC data, of which, the GLASS-GLC used satellite multi-source fusion approach and MODIS used direct MODIS sensor to derived their respective LUC inventories. The GLASS-GLC dataset spanning 1982-2015 but MODIS data is only available from 2000 onward. So, we replaced the GLASS-GLC by MODIS LULC data from 2002 to 2010 in the OSCAR model. The figure below shows annual fluctuations of the OSCAR simulated annual RF under global forestland changes using GLASS-GLS and MODIS from 2002 to 2010, respectively. Both RF results show annual fluctuations, though the RFs from the CLASS-GLC illustrate somewhat stronger oscillations. However, during this period, accumulated RFs subject to the global forestland changes driven by GLASS-GLC and MODIS LUC are 0.0165 Wm$^{-2}$ and 0.0157 W m$^{-2}$, respectively, indicating only a 5% difference between the two satellite remote sensing derived LUC datasets.

Sun et al. (2022) compared the applications of six LULC products in the identification of LUCs in Northwestern China. Their results indicated, while the GLASS-GLC and MODIS (MCD-12Q1) were not superior to other four products (developed only for China), these two datasets were of most temporal and spatial consistency. This paper has been cited in the revised paper.

These discussions have been summarized in a new paragraph in section 2.2 (third paragraph).

[Figure]

The GLASS-GLC dataset was further compared temporally and spatially with the LUH1 dataset in Supplementary Figures S1 and S2. While the GLASS-GLC is superior to the LUH1, the magnitude of GLASS-GLC is comparable to LUH1 dataset. Eq. S1 defines the principle of the OSCAR model to predict ARF, which is closely related to the area of LUC, and therefore, the fluctuation of the ARF results is also reflected by the land use conversion data of the dataset, which is well reflected by Figure 3 and Figs. S3-S13 of this paper.

The result has been added to the revised Supplementary Text 2 (the last paragraph).

Sun, W. et al. Land use and cover changes on the Loess Plateau: A comparison of six global or national land use and cover datasets. Land Use Policy 119, 106165 (2022).

*7. The authors mentioned that OSCAR does not estimate surface albedos itself. Instead, it collected surface albedos in different countries and regions from literature and other climate models. However, as I know, surface albedo shows large spatial variation even for the same land types. Please provide a direct evaluation of the OSCAR surface albedo using the available remote sensing data, e.g., MODIS. Without such evaluation, the results from this study can be unreliable.*

**Response:** We thank the Reviewer's suggestion. We agree with the Reviewer that surface albedo vary spatially and temporarily. Albedo can change significantly over time due to seasonal effects, such as snow cover, vegetation growth, and land use changes. This temporal variability makes it challenging to obtain consistent albedo measurements. Surface albedo can also vary greatly over small spatial scales due to changes in land cover, vegetation type, soil moisture, and surface roughness. This spatial complexity complicates the estimation process, particularly in a global scale and a long-term perspective. As a result, it is difficult to obtain an "accurate albedo dataset". In fact, the albedo data from most widely used MODIS and GLASS is only available from 2000 onward, whereas our model simulations extend from 1982 to 2010. Some satellite instruments lunched in the 1980s could provide albedo data in the early stage but these data seemed not consistent with MODIS data.

This study aims to demonstrate the responses of RF to the tempo-spatial resolution of LUC. Following the Reviewer's main comment 4, we compared modeled RF using MODIS LUC and GLASS-GLC datasets from 2002 to 2010, the results revealed minor differences. However, we do recognized uncertainties in OSCAR albedo data in the revised section 2.1.

Nevertheless, efforts will be made in future to replace OSCAR albedo data by satellite remote sensing albedo data, but this will be a time-consuming and heavy task.

**Minor concerns:**

*1. L136-137: Please provide the citation.*

**Response:** Done, thanks!

*2. Data availability: Please also share the model output in the study.*

 **Response:** Done, thanks! We provide a repository to access relevant output data (https://doi.org/10.5281/zenodo.14586249).

**Below are my comments from the previous round of review.**

**Response:** Thanks! These comments have been addressed before the discussion stage.

*Land use change has a large impact on global climate change. This study quantified the LUC-induced albedo change and its radiative forcing based on remote sensing-derived LUC dataset. The study is interesting and the results and conclusions are meaningful. However, some issues are needed to be carefully revised: 1) The authors set a lot of thresholds when calculating RFs and carrying out the sensitivity analysis, without accounting the corresponding reasons. 2) The urban change is not accounted for, which can induce large uncertainty. Please see below for my specific comments.*

*Major concerns:*

1. *Line 15-18: Land use change has complex impacts on climate change. Whether it is cooling or warming effect depends on the specific conversions from one land use to another land use. Land use change can emit GHG, change surface albedo and ET, and further affect climate. However, which factor dominates depends on the specific conditions.*

2. *L103: The authors assigned a 5% uncertainty in modeled ARF induced by LUC uncertainty. However, why the authors set this value is unclear. How did the authors use this in the model?*

3. *L131: Some studies (e.g., Ouyang et al., 2022) have shown that urbanization has an albedo-induced warming effect. However, this study neglected the urban change, which may induce large uncertainties.*

*Ouyang, Z., Sciusco, P., Jiao, T. et al. Albedo changes caused by future urbanization contribute to global warming. Nat Commun **13**, 3800 (2022). https://doi.org/10.1038/s41467-022-31558-z*

1. *L139: In the sensitivity experiments, why did the authors set this threshold of 20%?*

2. *GLASS LC data cover 1982-2015. Why did the authors just analyze the data from 1983-2010?*

3. *Section 2.1: Please clarify how OSCAR model uses the land use data, considering it is not spatially resolved.*

4. *3: why did the authors select 1% as the threshold?*

5. *Figure 1 & 2: Please explain why the simulations in S1 and S2 show very different trends in the global average and regional values. Please add the corresponding LUC analysis and clearly explain it in the main text.*

6. *Line 157: Considering that there is a big difference between LUH1 and GLASS, replacing LUH1 with GLASS in 1982 can induce some uncertainties. Please discuss it. How did the authors harmonize these two LUC datasets?*

7. *The work neglected the impacts of LUC on surface roughness, which deserves some discussion.*

8. *In the methods section, the authors mainly introduced the sensitivity analysis. Also need to introduce how to use two LUC datasets for the analysis of albedo-induced RFs. Please also introduce the objective of the sensitivity analysis. In the sensitivity analysis, the authors define multiple new variables. However, some of them are not easy to understand. Please make them easier to follow.*

9. *There is a large spatial variation of surface albedo. Surface albedo is dependent on the vegetation structure, leaf/soil albedo and surface topography. I am curious how OSCAR considers the spatial variation of surface albedo.*

*Minor concerns:*

1. *L67: Please provide the citation.*

2. *L163: to2010 -> to 2010.*

3. *L267: This equation can be moved to methods section.*

4. *Figure 3: Effective area and RF have different units. Why did the authors put them together?*